# *APOE* Genotypes Modulate Inflammation Independently of Their Effect on Lipid Metabolism

**DOI:** 10.3390/ijms232112947

**Published:** 2022-10-26

**Authors:** María Civeira-Marín, Ana Cenarro, Victoria Marco-Benedí, Ana M. Bea, Rocío Mateo-Gallego, Belén Moreno-Franco, José M. Ordovás, Martín Laclaustra, Fernando Civeira, Itziar Lamiquiz-Moneo

**Affiliations:** 1Lipid Unit, Hospital Universitario Miguel Servet, Instituto de Investigación Sanitaria Aragón (IIS Aragón), Centro de Investigación Biomédicas en Red Enfermedades Cardiovacsulares (CIBERCV), 50009 Zaragoza, Spain; 2Instituto Aragonés de Ciencias de la Salud (IACS), 50009 Zaragoza, Spain; 3Departamento de Medicina, Psiquiatría y Dermatología, Facultad de Medicina, Universidad de Zaragoza, 50009 Zaragoza, Spain; 4Departamento de Fisiatría y Enfermería, Facultad de Ciencias de la Salud y del Deporte, Universidad de Zaragoza, 22001 Huesca, Spain; 5Departamento de Medicina, Microbiología, Pediatría, Radiología y Salud Pública, Facultad de Medicina, Universidad de Zaragoza, 50009 Zaragoza, Spain; 6Nutrition and Genomics Laboratory, JM-USDA Human Nutrition Research Center on Aging, Tufts University, Boston, MA 02111, USA; 7Precision Nutrition and Obesity Program, IMDEA Alimentación, 28049 Madrid, Spain; 8Departamento de Anatomía e Histología Humanas, Facultad de Medicina, Universidad de Zaragoza, 50009 Zaragoza, Spain

**Keywords:** C-reactive protein, *APOE* genotype, lipoprotein subclasses, GlycA

## Abstract

The association between *APOE* genotypes and cardiovascular disease (CVD) is partially mediated by LDL-cholesterol concentration but persists after adjusting for lipid levels and other cardiovascular risk factors. Data from the Aragon Workers Health Study (AWHS) (n = 4159) and the Lipid Unit at the Hospital Universitario Miguel Servet (HUMS) (n = 3705) were used to investigate the relationship between C-reactive protein (CRP) levels and *APOE* genotype. Lipoprotein particle and GlycA concentrations were analyzed in a subsample from AWHS. *APOE* genotyping was carried out by the Sanger method in both cohorts. *APOE4* carriers had significantly lower levels of CRP than *APOE3* carriers. Furthermore, *APOE*4 carriers had cholesterol-enriched LDL particles compared to *APOE2* carriers. *APOE4* carriers also had higher concentrations of small, medium, and large LDL particles. CRP levels were not associated with lipoprotein particle number, size, or composition. GlycA levels were not associated with *APOE* genotypes. However, GlycA levels were significantly associated with the size and the amount of cholesterol contained in HDL, VLDL, and LDL particles. *APOE* genotype influences CRP concentration regardless of lipid profile. *APOE2* carriers showed the highest CRP levels, followed by *APOE3* and *APOE4*. A more atherogenic lipid profile, but not inflammatory markers could partly explain the higher CVD risk observed in *APOE4* carriers.

## 1. Introduction

Apolipoprotein E (ApoE) is a structural component of chylomicrons, as well as very-low-density (VLDL) and high-density (HDL) lipoproteins, which mediates the clearance of triglyceride-rich remnants, participates in the flow of cholesterol in macrophages in tissues, and modulates the generation of new HDL particles [1,2]. Thus, it plays a central function in lipid metabolism. ApoE is codified by the *APOE* gene, located in chromosome 19, q13.32, constituting three common alleles (E2, E3, and E4). The most common ApoE isoform is ApoE3, which has cysteine at residue 130 (SNV rs429358) and arginine at residue 176. ApoE2 has cysteine at residues 130 (SNV rs429358) and 176 (SNV rs7412). In contrast, the less common isoform, ApoE4, has arginine at both residues [3,4]. The receptor-binding function and lipid affinity of ApoE is particular to each isoform. Consequently, individuals carrying the ApoE4 isoform have higher low-density lipoprotein (LDL) cholesterol than those carrying the ApoE2 allele, whose binding capability to the LDL receptor is less than 2% [5]. ApoE3 is considered the wildtype reference isoform. In addition, ApoE2 has been associated with dysbetalipoproteinemia, while ApoE4 has been associated with Alzheimer’s disease and a higher risk of atherosclerosis and cardiovascular disease [2,4]. The association of the *APOE* genotype with cardiovascular disease is partially mediated by its influence on low-density lipoprotein (LDL) cholesterol concentrations. However, it persists despite adjustment for plasma lipids and classical cardiovascular risk factors [6].

One of the potential mechanisms for the association of the *APOE* locus with cardiovascular disease could be through inflammation-related mechanisms [7,8,9]. *APOE* gene variation has been associated with the concentration of C-reactive protein (CRP), a well-established marker of inflammation and an independent risk factor for cardiovascular disease (CVD) [10]. However, the mechanism of this association needs to be determined. In addition to CRP, a new biomarker of systemic inflammation, the glycoprotein acetyls (GlycA), was recently discovered for early cardiovascular risk in both young and older people [11,12]. In addition to knowing the central role of *APOE* in lipid metabolism, we hypothesize that this relationship between the *APOE* genotype and markers of inflammation could be related to lipid metabolism. Therefore, the objective of this study was to investigate the association of the *APOE* locus with CRP levels in two cohorts: patients from a Lipid Unit undergoing regular follow-up and subjects from the general population. Furthermore, this work analyzes the relationship between CRP and *APOE* genotype, which may depend on the lipid profile, by studying the composition and quantity of lipoparticles, as well as other markers of inflammation, such as GlycA, in a large group of individuals from the general population.

## 2. Results

Data from 7864 subjects were collected after applying exclusion criteria from both cohorts, including 3705 subjects from Lipid Unit from Hospital Universitario Miguel Servet (HUMS) and 4159 subjects from Aragon Workers’ Health Study (AWHS). As expected, HUMS subjects had significantly higher levels of total and LDL-cholesterol, TG, ApoA1, ApoB, and Lp(a) and lower HDL-cholesterol levels than the AWHS general population. HUMS subjects were significantly younger and had a significantly higher prevalence of diabetes. AWHS subjects were predominantly men (94.3%) in contrast to HUMS (49.4%). The APOE genotype distribution was significantly different between cohorts. The HUMS cohort had a higher prevalence of APOE2 and APOE4 alleles than the AWHS cohort (Table 1).

Clinical and biochemical characteristics according to APOE genotype are summarized in Table 2. As expected, levels of TC, TG, ApoA1, and ApoB were significantly different according to the APOE genotype. APOE4 carriers had higher total and LDL-cholesterol values and lower TG levels (*p* < 0.001 in all cases). The Lp(a) levels were significantly different according to APOE genotype: higher in APOE4 carriers than APOE3 carriers and APOE2, with the lowest Lp(a) being observed in ApoE2/E2 subjects (*p* < 0.001 and *p* = 0.036, 4159, and 3705 for AWHS and HUMS, respectively, Table 2).

### 2.1. APOE Genotype and C-Reactive Protein

Table 3 shows CRP levels according to APOE genotype in HUMS and AWHS cohorts. Both cohorts showed that APOE3/E3 subjects had greater levels of CRP than APOE3/E4 and APOE4/E4, respectively, (*p* < 0.001, in both cohorts), while higher values of CRP were found in E2 carriers.

CRP levels were inverse weakly correlated to HDL-cholesterol levels and directly correlated to TG levels. Moreover, in the AWHS cohort, we found a positive weak correlation of LDL-cholesterol and TC levels with CRP (Figure 1).

### 2.2. Association of APOE Genotype with Lipoprotein Particle Number, Composition, and Size

Among participants from the AWHS cohort, 1128 random serum samples were analyzed by NMR. A total of 132 subjects were carriers of at least one APOE2 allele, including APOE2/E2 and APOE2/E3 genotypes, 813 subjects were APOE3/E3, and 183 subjects were APOE3/E4 or APOE4/E4. HDL and VLDL lipoprotein subclasses did not show significant differences according to the APOE genotype (Appendix A). Nevertheless, LDL composition was significantly different according to APOE genotype. Subjects carrying APOE4 alleles presented a higher amount of cholesterol within LDL than carriers of the APOE2 allele, including small, medium, and large LDL particles (*p* = 0.004, *p* = 0.003, *p* = 0.003, and *p* = 0.005, respectively). APOE4 carriers also had a higher concentration of the small, medium, and large LDL particles (*p* = 0.006, *p* = 0.004, and *p* = 0.008, respectively, Table 4). These differences in LDL composition were maintained when subjects were distributed into five groups according to APOE genotype (Appendix A). However, HDL and VLDL subclasses did not show differences when subjects were distributed into five groups according to APOE genotype (Appendix A). 

### 2.3. Association of APOE Genotype with Inflammatory Markers

Table 5 shows CRP and GlycA levels according to APOE genotype. As seen above, APOE genotype influenced CRP concentration regardless of lipid profile, showing that APOE2 carriers had the highest CRP levels, followed by APOE3 and APOE4 carriers. This latter group had the lowest levels of CRP (*p* = 0.009, Table 5). Moreover, we found that CRP levels were associated with APOE genotype independently of any lipoprotein particle number size or composition, suggesting that the association between CRP and APOE genotype seems independent of the lipid profile (Table 5). GlycA did not show any association with the APOE genotype. However, GlycA showed a significant association with the size and the amount of cholesterol contained in HDL, VLDL, and LDL (*p* = 0.002, *p* = 0.002, *p* = 0.027, *p* < 0.001, *p* < 0.001, and *p* < 0.001, respectively). However, it must be taken into account that the AWHS cohort was predominantly male, and that these associations were confirmed in a population with a homogeneous distribution of both sexes. 

## 3. Discussion

This study showed that the CRP serum levels and *APOE* genotype are associated, and that the lipoprotein particle profile does not modulate this association. *APOE4* carriers have significantly higher plasma TC concentrations due to higher levels of all LDL particle subfractions and a higher cholesterol concentration in each one of them. Conversely, the *APOE* genotype does not seem related to GlycA concentrations. However, this systemic inflammatory biomarker was associated in our cohort with the size and the amount of cholesterol contained in VLDL, LDL, and HDL. 

The relationship between *APOE* genotype and CVD has been well established. In the Framingham Study, Lahoz et al. [13] concluded that the presence of the *APOE2* or *APOE4* alleles in men was associated with significantly greater CVD risk than *APOE3/3* carriers. More recently, Dankner et al. [14] and Weiss et al. [6] demonstrated that, in men, the presence of the *APOE4* allele was associated with CVD after adjusting for several confounding factors, such as age, ethnicity, physical activity, hypertension, diabetes, lipid levels, and lipid-lowering drugs. This association was also demonstrated in patients with type 2 diabetes and end-stage renal disease [15]. However, the origin of this association has not yet been completely elucidated. In a meta-analysis, Sofat et al. [16] concluded that there is no evidence of an association of circulating ApoE concentration with CVD events, indicating that isoform-specific functions may explain the established association between *APOE* genotype and CVD events rather than circulating concentrations of ApoE.

Two potential mechanisms behind the association of the *APOE* genotypes with CVD are the variation in lipid concentrations and the inflammation response to the different *APOE* alleles [17]. It is well established that *APOE4* carriers have higher LDL-cholesterol concentration, probably via downregulation of the LDL receptor due to a higher affinity of APOE4-rich VLDL particles. In our study, *APOE4* carriers had significantly higher cholesterol values and a higher number of all subclasses of LDL particles, associated with a more atherogenic lipid profile. 

Numerous studies have identified inflammatory biomarkers associated with the development of CVD, including CRP, TNF-α, and several interleukins, or dysfunctional biomarkers of adiposity tissue, such as leptin or adiponectin [18,19], which improve the CVD risk classification, especially CRP [20,21]. Moreover, some studies have suggested that the association between *APOE* and CVD risk is partially mediated by a significant impact on inflammatory biomarkers, such as CRP levels, additionally modulated by factors such as sex and adiposity [22]. 

CRP is a short pentraxin found mainly as a pentamer in circulation or as non-soluble CRP monomers (mCRP). It is synthesized in the liver and induced by proinflammatory stimuli, such as IL-6, IL-1β, and TNF-α. CRP has a role in the humoral innate immune response but also contributes to the progression of cardiovascular disease by recognizing and binding to multiple intrinsic ligands. Although mCRP is not present in the healthy vessel wall, it becomes detectable in the early stages of atherogenesis. It accumulates during the progression of atherosclerosis, contributing to plaque instability by increasing the expression of endothelial cell adhesion molecules [23]. CRP plasma levels are also determined by polymorphisms in the CRP gene. These polymorphisms also modulate CVD risk [24]. Previous studies have shown that CRP levels are associated with the *APOE* genotype, indicating that *APOE4* carriers have significantly lower and *APOE2* carriers have significantly higher values of CRP than *APOE3/E3* subjects [25,26]. This association is in contrast with the positive association between *APOE4* and CVD. However, Tziakas et al. reported that *APOE4* carriers had lower values of the atheroprotective IL-10, hypothesizing that perhaps this mechanism was the cause of the association between the *APOE* and CVD [27]. Another study reported significantly higher levels of lipoprotein-associated phospholipase A2, a vascular inflammation marker, in *APOE4* carriers [28]. Our study found the same previously described relationship between CRP and *APOE* genotype. In addition, we found that this relationship seems to be independent of the lipid profile. 

GlycA represents the integrated concentration and glycosylation of numerous acute-phase proteins (predominantly α-1-acid glycoprotein, haptoglobin, and α-1-antitrypsin) released in the inflammatory state. This novel inflammatory biomarker has a positive predictive value for the future development of type 2 diabetes and CVD independently of CRP in different cohorts [12,29], especially in young individuals [11]. GlycA was consistently associated with impaired endothelial function and increased diastolic blood pressure, suggesting a link between chronic inflammatory processes. GlycA is also associated with lifestyle, including physical activity, smoking habits, or socioeconomic status, and metabolic disorders [11]. We found that GlycA is significantly positively associated with the amount of cholesterol in VLDL, LDL, and HDL and the number of all these particles. No previous studies have reported the relationship between GlycA and APOE genotype. Our study found that the *APOE* genotype seems unrelated to the glycoprotein acetyl levels, suggesting that *APOE4* carriers present a high risk of CVD due to a pathogenic mechanism unrelated to GlycA. 

Our study had some limitations. First, our data were observational and do not imply causality. Second, 1% of the subjects from the HUMS were on lipid-lowering treatment at the time of the analysis, and it is known that reducing LDL-cholesterol with drugs, especially statins [30], reduces the concentration of CRP. However, the fact that the lipid-lowering treatment was homogeneously distributed among the different *APOE* genotypes and that the results were replicated in the AWHS cohort makes it unlikely that this fact had a relevant influence on the results. Thirdly, the analysis carried out generating small groups based on the *APOE* genotype may mean that the findings found are not generalizable. However, for this very reason, we also carried out the distribution of the individuals according to the *APOE* allele, observing that the relationship between the *APOE* genotype and CRP was also maintained, while it did not exist in the case of GlycA. Lastly, the AHWS cohort was predominantly male, which means that the results could have been partially biased. However, the HUMS cohort had a fairly homogeneous distribution between women and men, and we obtained the same results in both regarding the relationship between *APOE* genotype and CRP values. In addition, in both cases, the association studied was adjusted for age, sex, and BMI.

## 4. Materials and Methods

### 4.1. Design

This research was a cross-sectional, observational study that used data from two sources: the Aragon Workers’ Health Study (AWHS) [31] and the Hospital Universitario Miguel Servet (HUMS), both in Zaragoza (Spain).

### 4.2. Participants

#### 4.2.1. AWHS

The AWHS is a longitudinal Caucasian cohort study started in 2009 based on an automobile assembly plant in Zaragoza, Spain. AWHS involves annual medical examinations and biological samples [31]. All workers were offered to participate in the study, and the response rate was 94.5%. The sample is predominantly male (>95%). Exclusion criteria included a history of personal CVD or the presence of clinical conditions that limited survival to less than 3 years. 

#### 4.2.2. HUMS

All consecutive unrelated studied patients with hyperlipidemia aged 18 to 80 from the Lipid Unit of HUMS from January 2006 to July 2022 were recruited for lipid research. This Lipid Unit is located in Zaragoza (Spain), and practically all of the individuals that comprise it are Caucasian. Subjects were excluded for the presence of secondary causes, including nephrotic syndrome, uncontrolled hypothyroidism (thyroid-stimulating hormone >6 mU/L), cholestasis (direct bilirubin >1 mg/dL), high alcohol intake (>30 g/day), or use of drugs that promote disorders of lipid metabolism (anabolic steroids, protease inhibitors, cyclosporine, mTOR kinase inhibitors, or cyclophosphamide). Most HUMS patients were removed from their lipid-lowering medication before their first visit to the Lipid Unit. After 6 weeks without medication, a complete biochemical analysis was performed, except for subjects with prior CVD or very high cardiovascular risk.

All AWHS participants and HUMS patients included in the study followed a complete clinical exam, analytic laboratory tests, and *APOE* genotyping. Exclusion criteria included a prior history of CVD, acute infectious or inflammatory disease, chronic anti-inflammatory drug use, or CRP serum levels higher than 10 mg/L.

### 4.3. Biochemical Analysis

AWHS and HUMS used the same biochemical protocols. Blood samples were drawn after overnight fasting without lipid-lowering treatment for at least 5 weeks, except in patients from HUMS with very high CVD risk. Laboratory tests were performed on the same day as blood sampling. 

Serum glucose, triglycerides (TG), total cholesterol (TC), and HDL-cholesterol were measured by spectrophotometry (AU5800 Beckman Coulter Inc), and ApoAI and ApoB were measured by kinetic nephelometry (Immunochemistry Analyzer IMMAGE 800, Beckman Coulter, Brea, CA, USA). Lipoprotein(a) (Lp(a)) concentration was measured by rate nephelometry using LPAX reagent in conjunction with IMMAGE Immunochemistry Systems and Lp(a) Calibrator (OMS/IFCC SRM 2 B) (Beckman Coulter), following the manufacturer’s instructions. LDL-cholesterol was calculated using the Friedewald equation. Whole-blood HbA1c was measured by reverse-phase cationic exchange chromatography with double wave-length colorimetry quantification (Analyzer ADAMS A1c HA-810, Arkray Factory). High-sensitivity CRP was measured by latex-enhanced immunoturbidimetry (DXC 700 Au, Beckman Coulter).

### 4.4. APOE Genotype

Genomic DNA from whole-blood samples was isolated using standard methods. Exon 4 of the APOE gene was amplified by polymerase chain reaction and purified by ExoSap-IT (USB), as previously described [32]. Amplified fragments were sequenced by the Sanger method using the BigDye 3.1 sequencing kit (Applied Biosystems, Waltham, MA, USA) in an automated ABI 3500xL sequencer (Applied Biosystems). DNA sequences were analyzed using Variant Reporter software (Applied Biosystems).

### 4.5. Lipoprotein Particles and GlycA Concentration

Lipoprotein particles and GlycA were quantified from serum samples in a subset of 1128 volunteers from AWHS using high-throughput nuclear magnetic resonance (NMR) metabolomics (Nightingale Health Ltd., Helsinki, Finland). This method provides simultaneous quantification of lipids and lipoprotein subclass profiling with lipid concentrations within 14 subclasses. Details of the experimentation and applications of the NMR metabolomics platform were described previously [33,34]. In this study, for each lipoprotein subclass in the LDL, VLDL, and HDL range (small, medium, and large LDL; very small, small, medium, large, very large, and extremely very large VLDL; small, medium, large, and very large HDL), we determined particle concentration (nmol/L), size, and cholesterol and TG concentration (mmol/L). 

### 4.6. Statistical Analysis

Continuous variables were expressed as the mean ± SD or median [25th percentile–75th percentile] as applicable, and categorical (nominal) variables were reported as percentages of the total sample. The *p*-value was calculated by ANOVA test or Kruskal–Wallis and chi-square tests, as appropriate. The relationship between CRP and *APOE* genotype was studied adjusted by age, sex, body mass index (BMI), TC, TG, ApoA1, and ApoB using linear regression models. The composition and fractions of VLDL, LDL, and HDL according to *APOE* alleles were adjusted by age, sex and BMI using linear regression models. All statistical analyses were performed with R version 3.5.0, including tidyverse, xlsx, dlpyr, gglot2, GGally, Hmsic, corrplot, and performanceAnalytics packages; significance was set at *p* < 0.05 [35].

## 5. Conclusions

In summary, we found a relationship between *APOE* genotypes and CRP plasma concentration in dyslipidemic subjects and the general population. *APOE4* carriers had significantly lower levels of CRP than *APOE3* carriers, who had lower levels than *APOE2* carriers independently of age, sex, and BMI. This relationship between CRP and *APOE* genotypes seems to be independent of the lipid profile. Therefore, the higher risk of CVD associated with the ApoE4 allele could be partly explained by the higher levels of cholesterol-enriched LDL particles. Lastly, we did not observe significant independent associations between GlycA and *APOE* genotypes in our cohorts.

## Figures and Tables

**Figure 1 ijms-23-12947-f001:**
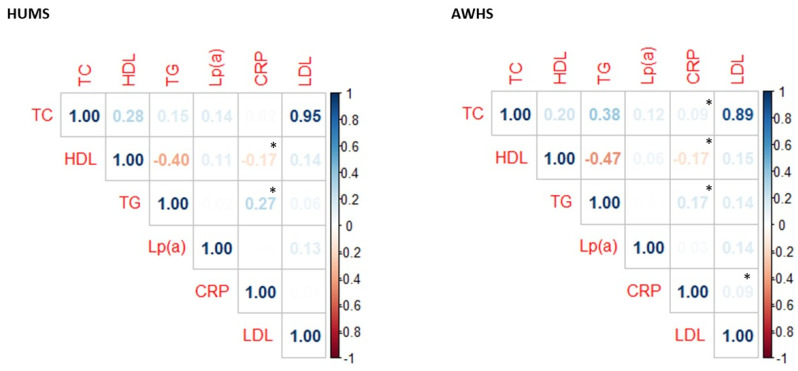
Correlation between CRP levels and lipid profile in AWHS and HUMS cohorts. * *p* < 0.05 calculated using the Spearman method. TC: total cholesterol, HDL: high-density lipoprotein cholesterol, TG: triglyceride, Lp(a): lipoprotein (a), CRP: C-reactive protein, LDL: low-density lipoprotein cholesterol.

**Table 1 ijms-23-12947-t001:** Clinical and biochemical characteristics in both cohorts.

		HUMS (N = 3705)	AWHS (N = 4159)	ES (95% CI)	*p*
Age, years		47.0 ± 14.5	48.2 ± 8.70	1.21 (0.69–1.73)	<0.001
Men, *n* (%)		1875 (49.4%)	3921 (94.3%)	NA	<0.001
BMI, kg/m^2^		26.4 ± 4.38	27.6 ± 3.81	1.19 (1.01–1.37)	<0.001
TC, mg/dL		281 ± 72.8	213 ± 37.8	−67.6 (−70.1–(−65.1))	<0.001
HDL-cholesterol, mg/dL		55.2 ± 17.6	53.1 ± 11.7	−2.16 (−2.81–(−1.50))	<0.001
LDL-cholesterol, mg/dL		194 ± 66.1	128 ± 42.5	66.0 (85.0–50.2)	<0.001
TG, mg/dL		132 (86.0–226)	119 (83.0–177)	13	<0.001
Apolipoprotein A1, mg/dL		154 ± 48.3	142 ± 22.3	−12.3 (−14.0–(−10.7))	<0.001
Apolipoprotein B, mg/dL		144 ± 46.5	106 ± 25.3	−37.9 (−39.5 –(−36.3))	<0.001
Lipoprotein(a), mg/dL		24.6 (8.90–64.1)	18.8 (9.00–44.0)	5.80	<0.001
Glucose, mg/dL		90.0 (83.0–99.0)	96.0 (88.0–104)	−6.00	<0.001
Hb1Ac, %		5.40 (5.20–5.70)	5.40 (5.20–5.60)	0.00	0.540
Hypertension, *n* (%)		716 (19.4%)	735 (17.8%)	NA	0.059
Diabetes, *n* (%)		191 (5.17%)	128 (3.11%)	NA	<0.001
*APOE* Genotype, *n* (%)	E2/E2	27 (0.73%)	17 (0.41%)	NA	<0.001
	E2/E3	277 (7.47%)	446 (10.7%)		
	E3/E3	2477 (66.9%)	2980 (71.6%)		
	E3/E4	847 (22.9%)	684 (16.4%)		
	E4/E4	77 (2.08%)	32 (0.77%)		

Quantitative variables are expressed as means ± standard deviations, except for variables not following a normal distribution that are expressed as medians (interquartile ranges). Qualitative variables are expressed as percentages. The *p*-value was calculated using the Student *t-*test or Mann–Whitney U and chi-square tests, as appropriate. The effect size (ES) and 95% confidence interval (CI) were calculated using a linear regression model. NA: not applied.

**Table 2 ijms-23-12947-t002:** Clinical and biochemical characteristics according to *APOE* genotype in both cohorts.

HUMS	ALL (N = 3705)	E2/E2 (N = 27)	E2/E3 (N = 277)	E3/E3 (N = 2477)	E3/E4 (N = 847)	E4/E4 (N = 77)	*p*
Age, years	47.0 ± 14.5	46.7 ± 13.3	49.4 ± 14.1	47.1 ± 14.6	46.7 ± 14.3	40.7 ± 14.3	<0.001
Men, *n* (%)	1875 (49.4%)	20 (74.1%)	168 (60.6%)	1224 (49.4%)	422 (49.8%)	41 (53.2%)	0.046
BMI, kg/m^2^	26.4 ± 4.38	27.7 ± 3.36	27.7 ± 4.03	26.3 ± 4.30	26.3 ± 4.64	26.5 ± 4.89	<0.001
TC, mg/dL	281 ± 72.8	361 ± 149	271 ± 95.3	281 ± 70.8	282 ± 66.5	280 ± 54.1	0.961
HDL-cholesterol, mg/dL	55.2 ± 17.6	53.4 ± 15.4	49.8 ± 15.6	55.8 ± 17.7	55.1 ± 17.5	54.9 ± 18.6	0.030
LDL-cholesterol, mg/dL	194 ± 66.1	153 ± 111	175 ± 70.5	195 ± 67.6	198 ± 59.8	197 ± 50.6	<0.001
TG, mg/dL	132 (86.0–226)	454 (298–705)	174 (103–311)	127 (84.0–215)	133 (89.0–227)	126 (100–222)	<0.001
Apolipoprotein A1, mg/dL	154 ± 48.3	151 ± 40.5	148 ± 30.5	155 ± 53.9	155 ± 34.3	151 ± 39.3	0.361
Apolipoprotein B, mg/dL	144 ± 46.5	106 ± 45.3	131 ± 40.7	144 ± 48.8	148 ± 40.2	152 ± 38.0	<0.001
Lipoprotein(a), mg/dL	24.6 (8.90–64.1)	19.4 (4.63–42.4)	21.7 (7.93–60.0)	25.4 (9.38–64.6)	23.9 (8.21–64.9)	17.0 (5.69–46.8)	<0.001
Glucose, mg/dL	90.0 (83.0–99.0)	87.0 (82.0–106)	91.0 (82.0–101)	90.0 (83.0–99.0)	90.0 (83.0–97.0)	88.0 (81.0–96.3)	0.538
Hb1Ac, %	5.40 (5.20–5.70)	5.50 (5.15–5.95)	5.50 (5.20–5.80)	5.40 (5.20–5.70)	5.40 (5.20–5.70)	5.40 (5.10–5.60)	0.162
Hypertension, *n* (%)	716 (19.4%)	7 (25.9%)	63 (22.9%)	483 (19.6%)	150 (17.8%)	13 (16.9%)	0.323
Diabetes, *n* (%)	191 (5.17%)	3 (11.1%)	20 (7.28%)	121 (4.90%)	44 (5.21%)	3 (3.90%)	0.281
**AHWS**	**ALL (N = 4159)**	**E2/E2** **(N = 17)**	**E2/E3 (N = 446)**	**E3/E3** **(N = 2980)**	**E3/E4** **(N = 684)**	**E4/E4** **(N = 32)**	** *p* **
Age, years	48.2 ± 8.70	47.9 ± 8.12	47.7 ± 9.26	48.4 ± 8.56	48.0 ±9.00	50.2 ± 7.58	0.428
Men, *n* (%)	3921 (94.3%)	17 (100%)	420 (94.2%)	2807 (94.2%)	645 (94.3%)	32 (100%)	0.554
BMI, kg/m^2^	27.6 ± 3.81	29.3 ±4.17	27.6 ± 3.83	27.5 ± 3.82	27.6 ± 3.78	27.8 ± 3.53	0.775
CT, mg/dL	213 ± 37.8	182 ± 33.2	202 ± 36.9	213 ± 37.4	219 ± 38.2	221 ± 41.7	<0.001
HDL-cholesterol, mg/dL	53.1 ± 11.7	49.5 ± 16.5	53.3 ± 11.4	53.3 ± 11.7	52.1 ± 11.6	49.2 ± 13.9	0.028
LDL-cholesterol, mg/dL	128 ± 42.5	123 ± 43.6	125 ± 43.9	128 ± 42.2	128 ± 42.8	122 ± 47.9	0.450
TG, mg/dL	119 (83.0–177)	182 (108–281)	135 (89.0–198)	117 (82.0–172)	119 (87.0–180)	150 (113–209)	<0.001
Apolipoprotein A1, mg/dL	142 ± 22.3	149 ± 26.4	144 ± 20.9	143 ± 22.6	140 ±21.7	141 ± 19.8	<0.001
Apolipoprotein B, mg/dL	106 ± 25.3	62.4 ± 20.2	94.5 ± 22.8	106 ±25.1	111 ± 24.7	115 ± 25.7	<0.001
Lipoprotein(a), mg/dL	18.8 (9.00–44.0)	8.00 (6.00–18.0)	17.0 (7.58–40.7)	18.5 (9.00–43.0)	20.0 (9.00–47.0)	25.0 (10.0–51.8)	0.036
Glucose, mg/dL	96.0 (88.0–104)	94.0 (89.0–101)	96.0 (88.0–104)	96.0 (88.0–104)	96.0 (88.0–104)	97.5 (88.8–114)	0.848
Hb1Ac, %	5.40 (5.20–5.60)	5.20 (4.80–5.50)	5.40 (5.20–5.60)	5.40 (5.20–5.60)	5.40 (5.20–5.60)	5.40 (5.25–5.85)	0.123
Hypertension, *n* (%)	735 (17.8%)	4 (23.5%)	68 (15.6%)	531 (18.0%)	123 (18.1%)	9 (28.1%)	0.360
Diabetes, *n* (%)	128 (3.11%)	1 (5.89%)	14 (3.20%)	89 (3.01%)	21 (3.91%)	3 (9.38%)	0.319

Quantitative variables are expressed as means ± standard deviations, except for variables not following a normal distribution that are expressed as medians (interquartile ranges). Qualitative variables are expressed as percentages. The *p*-value was calculated using the ANOVA test or Kruskal–Wallis and chi-square tests as appropriate, comparing variables according to APOE genotype. BMI: body mass index; TC: total cholesterol, HDL-cholesterol: high-density lipoprotein cholesterol, LDL-cholesterol: low-density lipoprotein cholesterol; TG: triglyceride.

**Table 3 ijms-23-12947-t003:** CRP levels according to *APOE* genotype in HUMS and AWHS cohorts.

HUMS	E2/E2 (N = 27)	E2/E3 (N = 277)	E3/E3 (N = 2477)	E3/E4 (N = 847)	E4/E4 (N = 77)	*p*
CRP, mg/L	2.09 ± 1.53	2.51 ± 1.02	2.32 ± 2.10	1.89 ± 1.89	1.64 ± 1.55	<0.001
**AHWS**	**E2/E2** **(N = 17)**	**E2/E3 (N = 446)**	**E3/E3** **(N = 2980)**	**E3/E4** **(N = 684)**	**E4/E4** **(N = 32)**	** *p* **
CRP, mg/L	2.51 ± 1.66	2.72 ± 2.07	2.63 ± 1.99	2.32 ± 1.93	1.43 ± 1.13	<0.001

Quantitative variables are expressed as means ± standard deviations. The *p*-value was adjusted by age, sex body mass index, total cholesterol, triglycerides, and apolipoproteins A1 and B. CRP: C-reactive protein.

**Table 4 ijms-23-12947-t004:** LDL composition and fractions according to *APOE* allele in a subsample from AWHS cohort.

	E2 Allele ^1^ N = 132	E3 Allele ^2^ N = 813	E4 Allele ^3^ N = 183	*p*
LDL size (nm)	23.56 ± 0.084	23.45 ± 0.075	23.53 ± 0.069	0.353
Cholesterol in LDL	1.548 ± 0.472	1.756 ± 0.486	1.843 ± 0.474	0.004
Triglycerides in LDL	0.185 ± 0.056	0.187 ± 0.052	0.192 ± 0.051	0.359
Small LDL particle (mol/L)	1.635 × 10^−7^ ± 4.212 × 10^−8^	1.799 × 10^−7^ ± 4.364 × 10^−8^	1.881 × 10^−7^ ± 4.134 × 10^−8^	0.006
Cholesterol in small LDL	0.280 ± 0.092	0.322 ± 0.092	0.338 ± 0.089	0.003
Triglycerides in small LDL	0.033 ± 0.0126	0.032 ± 0.010	0.034 ± 0.011	0.435
Medium LDL particle (mol/L)	1.387 × 10^−7^ ± 3.822 × 10^−8^	1.545 × 10^−7^ ± 3.974 × 10^−8^	1.620 × 10^−7^ ± 3.790 × 10^−8^	0.004
Cholesterol in medium LDL	0.456 ± 0.151	0.525 ± 0.154	0.553 ± 0.149	0.003
Triglycerides in medium LDL	0.050 ± 0.016	0.051 ± 0.014	0.052 ± 0.014	0.162
Large LDL particle (mol/L)	1.724 × 10^−7^ ± 4.326 × 10^−8^	1.893 × 10^−7^ ± 4.576 × 10^−8^	1.973 × 10^−7^	0.008
Cholesterol in large LDL	0.810 ± 0.233	0.908 ± 0.242	0.952 ± 0.237	0.005
Triglycerides in large LDL	0.101 ± 0.029	0.103 ± 0.028	0.106 ± 0.027	0.299

Quantitative variables were expressed as means ± standard deviations. The *p*-value was adjusted by age, sex, and BMI. ^1^ The E2 allele included subjects with the E2/2 and E2/3 APOE genotype. ^2^ The E3 allele included only subjects with the E3/3 APOE genotype. ^3^ The E4 allele included subjects with the E3/4 and E4/4 APOE genotype. LDL: low-density lipoprotein.

**Table 5 ijms-23-12947-t005:** CRP and GlycA levels according to APOE genotype in in a subsample from AWHS cohort.

	E2 Allele ^1^ N = 132	E3 Allele^2^ N = 813	E4 Allele ^3^ N = 183	*p*
CRP, mg/L	4.34 ± 5.59	3.66 ± 4.38	2.61 ± 2.91	0.009
Glycoprotein acetyls, mmol/L	1.52 ± 0.386	1.45 ± 0.287	1.48 ± 0.302	0.737

Quantitative variables are expressed as means ± standard deviations. The *p*-value was adjusted by age, sex BMI, HDL-, LDL-, and VLDL-size, and cholesterol in HDL, LDL, and VLDL. ^1^ The E2 allele included subjects with the E2/2 and E2/3 APOE genotype. ^2^ The E3 allele included only subjects with the E3/3 APOE genotype. ^3^ The E4 allele included subjects with the E3/4 and E4/4 APOE genotype. CRP: C-reactive protein.

## Data Availability

The data used to carry out this study can be provided upon request to the principal investigators.

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
