# Peer review of "APOE Genotypes Modulate Inflammation Independently of Their Effect on Lipid Metabolism"

_ijms, 2022, doi:10.3390/ijms232112947_

Round 1

Reviewer 1 Report

In the current manuscript, Civeira-Marín and co-workers present a comparison of lipid levels according to the ApoE genotype and evaluate their association with markers of neuroinflammation. The experimental settings are easy to follow. However some concerns exist that hinder the publication of the manuscript in its current form.

1.       The first sentence, reads incomplete. Perhaps a “which” is missing before “mediates”.

2.       Within the introduction a clear hypothesis is missing regarding the use of lipoprotein particles.

3.       Also, a clear hypothesis for the use of GlycA.

4.       In all tables, to have the statistic not only p-values is needed. Even, effect size is recommended.

5.       CVD, HUMS, AWHS have to be spelled out at their first appearance.

6.       Perhaps I missed something, but I found counterintuitive the organization of the manuscript: intro, results, methods, conclusions. There is important information in the methods that needs to be delivered before the results. E.g., diagnoses ad inclusion/exclusion criteria.

7.       Mention of the specific R packages used would be great to have.

8.       Where the implemented regression linear or non-linear?

9.       Also, please add all appropriate abbreviations to the tables.

10.   In the AHWS cohort the highest LDL and APP-A1 levels are found for E2/E2 carriers. How do the authors explain this? Also the lowest levels of APP B and lipoprotein were seen in this group.

11.   In the HUMS cohort, the highest TG are in E2/E2.

12.   A small discussion on the sample sizes shall be made. For example, even when not significantly different, in HUMS, the highest prevalence of Diabetes and hypertension was in E2/E2 carriers. However, this is not reflected on the glucose levels. Also, in AHWS E2/E2 are only barely overcome by E4/E4 carriers. Thus, it is doubtable that for groups with small sample sizes the findings are generalizable, but rather just and effect of the sample sizes/selection in the current study.

13.   Regarding the possible neuroprotective role that E2 is supposed to confer. How do the authors interpret the prevalence in the previous comment?

14.   In both cohorts, the highest levels of CRP are found in E3/E3 and the lowest in E4/E4. But the authors state in the intro that ApoE may act through or is associated with CRP.

15.   One possibility to improve the statistical approaches, overcoming the shortcomings of the study design, would be to have a match-group analysis. Where the N’s are matched or at least shorten, also by sex, age, BMI, and other possible factors to that of the smallest group. This would allow to have a more balanced design and statistical power. Including for comparisons between cohorts.

Author Response

Reviewer 1

In the current manuscript, Civeira-Marín and co-workers present a comparison of lipid levels according to the ApoE genotype and evaluate their association with markers of neuroinflammation. The experimental settings are easy to follow. However, some concerns exist that hinder the publication of the manuscript in its current form.

  1. The first sentence, reads incomplete. Perhaps a “which” is missing before “mediates”.

We agree with re review and following their indications, we have included “which” before mediate. In the current version, it reads: “Apolipoprotein E (ApoE) is a structural component of chylomicrons, very low-density (VLDL) and high-density (HDL) lipoproteins, which mediates the clearance of triglyceride-rich remnants, participates in the flow of cholesterol in macrophages in tis-sues and modulates the generation of new HDL particles [1,2]”

  1. Within the introduction a clear hypothesis is missing regarding the use of lipoprotein particles.

Following the reviewer’s indications, we have added the next sentence in the current manuscript version: “In addition to knowing the central role of APOE in lipid metabolism, we hypothesize that this relationship between the APOE genotype and markers of inflammation could be related to lipid metabolism.”

  1. Also, a clear hypothesis for the use of GlycA.

We agree with the reviewer. We have added the next sentence: “Besides to CRP, recently a new biomarker of systemic inflammation, the glycoprotein acetyls (GlycA), has been discovered novel inflammatory biomarker of early cardiovascular risk in both young and older people [11,12]”

  1. In all tables, to have the statistic not only p-values is needed. Even, effect size is recommended.

Following the reviewer suggestion, we have included the effect size and Coefficient Interval (CI) 95% in Table 1, which was calculated by linear regression model in all variables which following normal distribution. In the case of quantitative variables did not follow normal distribution, we calculated the Effect size as the difference from the mean. Only we have the possibility to calculate the effect size in Table 1, since in tables 2, 3 and 4 there are more than two groups in the independent variable.

  1. CVD, HUMS, AWHS have to be spelled out at their first appearance.

We have check all these terms and we have defined all these terms at their first appearance both in abstract and in the manuscript.

  1. Perhaps I missed something, but I found counterintuitive the organization of the manuscript: intro, results, methods, conclusions. There is important information in the methods that needs to be delivered before the results. E.g., diagnoses ad inclusion/exclusion criteria.

We agree with the reviewer. However, we used the word template facilitated by the submit system. And in this word was indicated that material and methods should be appeared at the end of manuscript.

  1. Mention of the specific R packages used would be great to have

Following the reviewer’s suggestion, we have included all package that we have used in these analysis. In the current version, it reads: “All statistical analyses were performed with R version 3.5.0, including tidyverse, xlsx, dlpyr, gglot2, GGally, Hmsic, corrplot and performance Analytics packages and significance was set at p<0.05 [15].”

  1. Where the implemented regression linear or non-linear?

The linear regression was implemented to analysis the relationship between CRP and APOE genotype adjusted by age, sex, BMI, total cholesterol, triglycerides and apolipoproteins A1 and B. Besides, we have used linear regression model to analyze the relationship between composition and fractions of HDL, VLDL and LDL according to APOE allele.

However, as it seems that this point is not clear, we have written this sentence from the statistics section of the material and methods section. In the current version, it reads: “The relationship between CRP and APOE genotype was studied adjusted by age, sex, body mass index (BMI), TC, TG, ApoA1 and ApoB using linear regression models. The composition and fractions of VLDL, LDL and HDL according to APOE alleles were studied adjusted by age, sex and BMI using linear regression models”

  1. Also, please add all appropriate abbreviations to the tables.

We have checked all abbreviation in the footnotes and we have added the next information in the Table 2: “LDL: Low-density lipoprotein” and the information in the footnote of Table 3: “CRP: C-reactive protein”

  1. In the AHWS cohort the highest LDL and APP-A1 levels are found for E2/E2 carriers. How do the authors explain this? Also the lowest levels of APP B and lipoprotein were seen in this group.

Firstly, we would like to apologize. Checking the original data, we have discovered that we have a mistake in table 3 in the AWHS cohort. We had a line dance and the line of 53.1 ± 11.7 corresponded to HDL cholesterol, 128 ± 42.5 corresponded to LDL cholesterol and 119 (83.0 – 177) are the correct values of TG. Therefore, we have found that E2/E2 carriers have lowest values of LDL cholesterol and apolipoproteins B and highest values of TG and apolipoproteins A1. E2/E2 carriers usually have a dysbetalipoproteinemia phenotype. This phenotype presents with high TG values, as well as very low apolipoprotein B values, which generates a very low ratio of non-HDL cholesterol/apolipoprotein B, which is characteristic of this disease (Christopher S Boot et al. Clin Chem 2019 Feb;65(2):313-320. doi: 10.1373/clinchem.2018.292425, Bibin Varghese et al. J Clin Lipidol. 2021 Jan-Feb;15(1):104-115.e9. doi: 10.1016/j.jacl.2020.09.011, Sniderman et al. J Clin Lipidol. 2007 Aug;1(4):256-63. doi: 10.1016/j.jacl.2007.07.006).

  1. In the HUMS cohort, the highest TG are in E2/E2.

As we have commented before, we have discovered a mistake in the Table 2 in the AWHS cohort. In the current version, we can see that E2/E2 have the highest values of TG as we expected.

  1. A small discussion on the sample sizes shall be made. For example, even when not significantly different, in HUMS, the highest prevalence of Diabetes and hypertension was in E2/E2 carriers. However, this is not reflected on the glucose levels. Also, in AHWS E2/E2 are only barely overcome by E4/E4 carriers. Thus, it is doubtable that for groups with small sample sizes the findings are generalizable, but rather just and effect of the sample sizes/selection in the current study.

We agree with the reviewer that distribution on the sample size could affect part of the result and we have this limitation in the discussion section. However, to try to avoid this error due to the small size of the groups, we grouped the subjects according to the APOE allele they carried to analyze the relationship between APOE and our variables of interest, which were CRP and GlyA. By distributing the groups in this way, we do see that there are significant associations after adjusting for BMI, sex, and age between CRP and APOE genotype. Despite this, we have included the next paragraph in the discussion section: “Thirdly, the analysis carried out generating small groups based on the APOE genotype, may mean that the findings found are not generalizable. However, for this very reason, we carried out the distribution of the individuals also according to the APOE allele, observing that the relationship between the APOE genotype and CRP is also maintained, while it does not exist in the case of GlycA.”

  1. Regarding the possible neuroprotective role that E2 is supposed to confer. How do the authors interpret the prevalence in the previous comment?

Although it is known that E4 individuals with the APOE gene have a higher risk of suffering from neurodegenerative pathologies such as Alzheimer's or Parkinson's. A recent study has shown that CSF IL-16 and IL-8 levels differed by sex and APOE genotype, with IL-16 being higher in female APOEɛ4 carriers compared to non-carriers, while that the opposite pattern was observed in men with IL-8. Furthermore, women had on average higher levels of CRP and ICAM1 in plasma, but lower levels of ICAM1, IL-8, IL-16 and IgA in CSF than men. In addition, the authors also indicated that carrying APOEɛ4 alleles and presenting a diagnosis of Alzheimer's or Parkinson's decreased plasma CRP concentration in both sexes (Paula Duarte-Guterman et al J Alzheimers Dis 2020;78(2):627-641. doi: 10.3233/JAD-200982). Therefore, although it is not our line of research and we are not experts in the field, it seems that the bibliography consulted indicates that it would not be CRP that could be behind the neurodegenerative effect associated with the E4 allele, but perhaps other markers of inflammation.

  1. In both cohorts, the highest levels of CRP are found in E3/E3 and the lowest in E4/E4. But the authors state in the intro that ApoE may act through or is associated with CRP.

We have found the highest levels of CRP are found in E2 allele carriers than in E3 allele carriers and E4 allele carriers (Table 5). In the introduction we indicated that: “APOE gene variation has been associated with the concentration of C-reactive protein (CRP), a well-established marker of inflammation and an independent risk factor for cardiovascular disease (CVD) [10]. However, the mechanism of this association needs to be determined”. As we indicated in the discussion section, previous studies have shown that CRP levels are associated with the APOE genotype, indicating that APOE4 carriers have significantly lower and APOE2, higher values of CRP than APOE3/E3 subjects []. This association is in contrast with the positive association between APOE4 and CVD. However, Tziakas et al. reported that APOE4 carriers had lower values of the atheroprotective IL-10, hypothesizing that perhaps this mechanism was the cause of the association between the APOE and CVD [30]. Another study reported significantly higher levels of lipoprotein-associated phospholipase A2, a vascular inflammation marker, in APOE4 carriers [31].

  1. One possibility to improve the statistical approaches, overcoming the shortcomings of the study design, would be to have a match-group analysis. Where the N’s are matched or at least shorten, also by sex, age, BMI, and other possible factors to that of the smallest group. This would allow to have a more balanced design and statistical power. Including for comparisons between cohorts.

We agree with the reviewer, for this reason, we analyzed the relationship between composition and fractions each lipoprotein particle and the relationship of APOE with inflammatory markers, such as CRP and GlycA according to APOE allele. Since with this grouping, we could group the E2E2 subjects together with E2E3 in a single group, obtaining an N=138, compared to the 813 individuals carrying the E3 allele (including only the E3E3) and the 183 subjects carrying the e4 allele (including individuals E3E4 and E4E4). Therefore, studying the relationship between CRP and GlycA and APOE according to APOE allele, we found that there is a significant relationship between APOE and CRP after adjusting by age, sex and BMI.

Reviewer 2 Report

1) Because AWHS cohort is predominantly male, the HUMS cohort should be also predominantly male. It is unclear if female participants were excluded. If they were, it should be also mentioned in the text. If, they were not, then if there is a “sex as a biological variable” effect, then that could be skewing the data collected from the HUMS cohort.

2) Because the AWHS cohort is predominantly male, then I suggest emphasizing that the results from the study are predominantly associated with a male population. This should be mentioned up-front in the result section.

3) What was the demographic? This should also be taken into consideration. For example, the risk of AD associated with the APOE4 allele differs between Caucasian and African American populations.  

4) What are the strengths and weaknesses of comparing these two groups and the limitations of the study? Some limitations are mentioned in the discussion, but I think that "sex as a biological variable" is another one that should be considered.

Line 117: how were the samples selected? Were they all male?

Minor changes:

Line 27: Please change E3 to APOE3

Line 45: Please change “common isoforms have arginine at both residues (ApoE4)” to “the less common isoform, ApoE4, has arginine at both residues”.

Line 76: Please correct the Table number. I think the sentence is referring to Table 2.

Author Response

Reviewer 2

1) Because AWHS cohort is predominantly male, the HUMS cohort should be also predominantly male. It is unclear if female participants were excluded. If they were, it should be also mentioned in the text. If, they were not, then if there is a “sex as a biological variable” effect, then that could be skewing the data collected from the HUMS cohort.

Although AWHS cohort is predominantly male, in the HUMS female participants were not excluded. However, we studied the association between CRP, GlycA and composition and fractions of each lipoprotein particle adjusted by age, sex and BMI. Besides, we have comment this possible limitation in the discussion section. In the current version, it reads: “Finally, the AHWS cohort is predominantly male, which could mean that the results could be partially biased. However, the HUMS cohort has a fairly homogeneous distribution between women and men and we obtained the same results in both regarding the relationship between APOE genotype and CRP values. In addition, in both cases, the association studied was adjusted for age, sex and BMI.”

2) Because the AWHS cohort is predominantly male, then I suggest emphasizing that the results from the study are predominantly associated with a male population. This should be mentioned up-front in the result section.

Following the reviewer suggestion, we have included the next sentence at the end results section: “However, it must be taken into account that the AWHS cohort is predominantly male, and that these associations are confirmed in a population with a homogeneous distribution of both sexes”. Besides, at the beginning of the results, in the description of the cohorts, it was already indicated that the AWHS population is predominantly male. In the current version, it reads: “AWHS subjects were predominantly men (94.3%) in contrast to HUMS (49.4%)”. In addition, I would like to comment that we have included this point as possible limitation in the discussion section.

3) What was the demographic? This should also be taken into consideration. For example, the risk of AD associated with the APOE4 allele differs between Caucasian and African American populations.  

Both cohorts are Caucasian. In fact, we have genotyped the AWHS recently and I can have confirmed that the ancestry analysis of AWHS cohort is really very similar to the European population. We have included this information in the Material and methods section. In the current version, it reads: “The AWHS is a longitudinal Caucasian cohort study started in 2009 based on an automobile assembly plant in Zaragoza, Spain” and “All consecutive unrelated studied patients with hyperlipidemia aged 18 to 80 from the Lipid Unit of HUMS from January 2006 to July 2022 were recruited for lipid research. This Lipid Unit are located in Zaragoza (Spain) and practically all of the individuals that comprise it are Caucasian”

4) What are the strengths and weaknesses of comparing these two groups and the limitations of the study? Some limitations are mentioned in the discussion, but I think that "sex as a biological variable" is another one that should be considered.

As weaknesses, selecting a sample from a Lipid Unit (from HUMS), can distort the results. In fact, we see that the distribution of APOE genotypes more associated with pathologies such as E2E2 or E4E4 are more frequent in the HUMS cohort. In addition to the fact that the lipid levels are clearly different, being much higher in the HUMS cohort, as expected. However, as strengths, we see that finding that there is an association between the APOE genotype and CRP in both cohorts, which reaffirms our idea that it is independent of the lipid profile.

On the other hand, we are aware that gender is a limiting factor in the case of the AWHS, our general population, however, we obtain the same results in the HUMS cohort, where the distribution of both genders is homogeneous and in addition to finding the significance after adjust for sex, age and BMI. Besides, we have included the next sentence at the end of results section, following the instructions indicated by first reviewer. In the current version, it reads: “However, it must be taken into account that the AWHS cohort is predominantly male, and that these associations are confirmed in a population with a homogeneous distribution of both sexes”. And we have commented this point in the discussion section too. In the current version, it reads: “Finally, the AHWS cohort is predominantly male, which could mean that the results could be partially biased. However, the HUMS cohort has a fairly homogeneous distribution between women and men and we obtained the same results in both regarding the relationship between APOE genotype and CRP values. In addition, in both cases, the association studied was adjusted for age, sex and BMI.”

Line 117: how were the samples selected? Were they all male?

The 1128 samples, which underwent NMR analysis, were randomly selected from the total of just over 5500 individuals that make up the entire AWHS cohort. Of the 1128 individuals, 1139 are male, so there is still a clear predominance of this sex, as expected. To clary this point, we have included the next sentence in the line 125: “Among participants from the AWHS cohort, 1128 random serum samples were analyzed by NMR”.

Minor changes:

Line 27: Please change E3 to APOE3

We have change E3 to APOE following the reviewer’s suggestion. In the current version, it reads: “APOE genotyping was carried out by the Sanger method in both cohorts. APOE4 carriers had significantly lower levels of CRP than APOE3 carriers.”

Line 45: Please change “common isoforms have arginine at both residues (ApoE4)” to “the less common isoform, ApoE4, has arginine at both residues”.

Following the reviewer’s suggestion, in the current version, it reads: “In contrast, the less common isoform, ApoE4, has arginine at both residues”

Line 76: Please correct the Table number. I think the sentence is referring to Table 2.

Table 1 is the correct cited. However, we have discovered that when we prepared the manuscript in the Word version provided, we did not include the complete Table 1 and we had forgotten the distribution of the APOE genotype comparing both cohorts. Therefore, we have added the APOE genotype in both cohorts.

Round 2

Reviewer 2 Report

Dear authors,

Lines 45-47. The sentence “In contrast, the less common isoform, ApoE4, has arginine at both residues or cysteine at residues 130 47 (SNV rs429358) and 176 (SNV rs7412) (ApoE2) (3,4).“ is not clear. Please revise.
